# A Novel Step-T-Junction Microchannel for the Cell Encapsulation in Monodisperse Alginate-Gelatin Microspheres of Varying Mechanical Properties at High Throughput

**DOI:** 10.3390/bios12080659

**Published:** 2022-08-19

**Authors:** Si Da Ling, Zhiqiang Liu, Wenjun Ma, Zhuo Chen, Yanan Du, Jianhong Xu

**Affiliations:** 1The State Key Laboratory of Chemical Engineering, Department of Chemical Engineering, Tsinghua University, Beijing 100084, China,; 2Department of Biomedical Engineering, School of Medicine, Tsinghua-Peking Center for Life Sciences Tsinghua University, Beijing 100084, China

**Keywords:** microdroplet, microfluidics, modified step-T-junction, viscoelasticity, cell encapsulation

## Abstract

Cell encapsulation has been widely employed in cell therapy, characterization, and analysis, as well as many other biomedical applications. While droplet-based microfluidic technology is advantageous in cell microencapsulation because of its modularity, controllability, mild conditions, and easy operation when compared to other state-of-art methods, it faces the dilemma between high throughput and monodispersity of generated cell-laden microdroplets. In addition, the lack of a biocompatible method of de-emulsification transferring cell-laden hydrogel from cytotoxic oil phase into cell culture medium also hurtles the practical application of microfluidic technology. Here, a novel step-T-junction microchannel was employed to encapsulate cells into monodisperse microspheres at the high-throughput jetting regime. An alginate–gelatin co-polymer system was employed to enable the microfluidic-based fabrication of cell-laden microgels with mild cross-linking conditions and great biocompatibility, notably for the process of de-emulsification. The mechanical properties of alginate-gelatin hydrogel, e.g., stiffness, stress–relaxation, and viscoelasticity, are fully adjustable in offering a 3D biomechanical microenvironment that is optimal for the specific encapsulated cell type. Finally, the encapsulation of HepG2 cells into monodisperse alginate–gelatin microgels with the novel microfluidic system and the subsequent cultivation proved the maintenance of the long-term viability, proliferation, and functionalities of encapsulated cells, indicating the promising potential of the as-designed system in tissue engineering and regenerative medicine.

## 1. Introduction

Cell encapsulation and cultivation within three-dimensional biomaterial scaffolds showcase great advantages over conventional 2D culture by providing a 3D physiochemical microenvironment similar to the in vivo physiological microenvironment [1,2,3,4]. In addition, cell-laden scaffold offers physical barrier protecting cells against external shear force and potential immune response, making it suitable for clinical in vivo usages [5,6,7,8]. Recently, cell-laden hydrogels have been widely employed in numerous applications, both academia and industry, including tissue engineering [9], biological analysis [10,11,12,13], drug delivery [14], cell therapy [15], pharmaceutical research [16], and regenerative medicine [17,18,19,20]. Nonetheless, cell macroencapsulation in bulk hydrogel suffers from poor nutrient, waste, and oxygen exchanges due to long transport distance. This would inherently lead to the undesired necrosis of cells located at the center of hydrogel, restricted intercellular communication, and inefficient manipulation over cell fate [21,22]. In contrast, because of its high surface-to-volume ratio, cell microencapsulation demonstrates its superiority in highly efficient heat and mass transfer; high responsiveness to external stimulations; and most importantly the compartmentalization of individual cells into isolated reactors easy to monitor, culture, and manipulate [13,23,24,25,26,27,28,29,30].

Common methods for cell microencapsulation that have emerged over past years include flow lithography, micro-molding, bulk emulsification, spray drying, and photolithography [31,32,33,34,35,36,37]. However, these methods are mostly limited by their low throughput, batch-to-batch differences, complexity in operation, possible cytotoxicity, and high cost [38,39,40]. On the other hand, microfluidic emulsification has drawn the attention from diverse avenues of research because of its precise control over the droplet morphologies and compositions, not to mention its high encapsulation efficiency, as well as the potential scalability via parallelization [41,42,43,44].

Several technical challenges still hamper the practical application of microfluidic technology, especially in the field of biomedical engineering. Among them all, the long-time exposure of cell-laden hydrogels to the cytotoxic oil continuous phase and surfactant urges the process of de-emulsification, which is usually time-/labor-consuming and may harm both the throughput of microencapsulation as well as the cell viability [40,45,46]. Numerous methods of de-emulsification have emerged over past years to transfer O/W cell-laden microdroplets from cytotoxic oil phase into aqueous cell culture medium. In principle, these methods rely on the pre-gelation, if not the complete gelation, of cell-laden microgels inside the oil liquid in the goal of ensuring the maximum throughput and maintaining the morphologies of generated microdroplets when transferring them across the oil–aqueous interface stabilized in the presence of surfactant. However, most of these methods demonstrate some degree of cytotoxicity and/or complexity in operation. For instance, some works directly merged alginate microdroplets with CaCl_2_ microdroplets inside microchannel via sophisticated microfluidic design, raising the concerns about the risk of microchannel clogging and the controllability over the merging process [7,47,48]. Other works reported the use of alternative calcium source such as CaCO_3_ and Ca-EDTA, releasing their calcium ions to cross-link alginate microdroplets upon contact with acid oil (pH < 5), which inevitably harms the cell viability [49,50]. Alternatively, photocurable biopolymers such as GelMA and PEGNB would also cross-link in the oil phase with the aid of a photoinitiator when exposing to UV light [51,52,53]. The exposure to UV light may raise the local temperature, harming cell viability and causing genetic mutation to encapsulated cells, not to mention the inherent cytotoxicity of most photoinitiators. Moreover, in order to avoid microchannel clogging, most of the previous works on microfluidic-based cell encapsulation employed rapidly cross-linking biopolymers such as alginate and PEG. Despite their biocompatibility, most of these biopolymers exhibit a rather linear elasticity [54]. Nonetheless, numerous recent studies reported the pivotal role of time-dependent viscoelasticity and stress–relaxation of biomaterials on cell fate and functionalities, suggesting the necessity of developing a viscoelastic biomaterial suitable for microfluidic technology [55,56,57,58,59]. In fact, the viscoelasticity is viewed as the material displaying increased deformation with time under the application of a defined force or stress. Cameron et al. reported the enhanced hMSC proliferation, spreading, and differentiation towards multiple cell lineage on highly viscoelastic hydrogels when compared to their corresponding purely stiff analogue [56]. Similarly, Bauer et al. found the enhanced spreading of myoblasts on stress–relaxing hydrogel substrate when compared to those cultured on purely elastic hydrogels of the same initial elastic modulus [55].

In this work, a modified step-T-junction microchannel with embedded capillary was employed to generate monodisperse cell-laden alginate–gelatin microspheres at high throughput where biocompatible soy bean oil with PGPR as surfactant was used as continuous liquid, as illustrated in Figure 1. The novel microchannel geometry would introduce a dual disturbance onto the dispersed liquid, allowing the generation of monodisperse microdroplets at high-throughput jetting regime as described in our previous work [60]. Using the natural properties of gelatin to freeze under coldness, the alginate–gelatin microspheres can be de-emulsified and transferred from oil phase into cell culture medium via centrifugation in a highly biocompatible and efficient manner while maintaining the morphologies of generated microspheres. The viscoelasticity, stress relaxation, stiffness, and water absorption of alginate–gelatin hydrogels at different alginate-to-gelation ratios, crosslinking times, and cross-linker concentrations were determined. The results demonstrated that the co-polymer network of ionically cross-linked alginate and covalently cross-linked gelatin exhibits a tunable mechanical performance over a wide range, which can provide the optimal biophysical 3D microenvironment with time-dependent viscoelasticity for encapsulated cells. Finally, HepG2 cells encapsulated within the alginate–gelatin microspheres via the novel microfluidic system maintained the long-term viability and cell functionality of urea secretion over a period of 6 days. In sum, the present work designed a novel microfluidic system allowing the high-throughput and greatly biocompatible generation of monodisperse alginate–gelatin microspheres with tunable viscoelastic mechanical performance over a wide range, whose applications can be further broadened in diversified avenues of biomedical engineering.

## 2. Experimental

### 2.1. Materials and Reagents

The alginate and the surfactant polyglycerol polyricinoleate (PGPR) were purchased from MACKLIN Biochemical Technology Co., Ltd. (Shanghai, China). The gelatin was purchased from Sigma-Aldrich (Shanghai, China). The soy bean oil was purchased from Jinlongyu Co., Ltd. All other chemicals were obtained from Aladin Co., Ltd. (Shanghai, China) unless specified.

### 2.2. Design and Fabrication of the Microfluidic Device

The microfluidic chip was fabricated using polymethylmethacrylate (PMMA) with a computerized numerical control (CNC) micromilling machine. The detailed design principle and fabrication of the novel step-T-junction microchannel with embedded microcapillary were described in our previous work [60]. Briefly, as shown in Figure 2, a stepped main microchannel for the introduction of the continuous oil liquid intersected with a perpendicular side microchannel, where a silica microcapillary was embedded for the flowing of dispersed alginate–gelatin precursor solution. The silica microcapillary has an outer diameter of 365 μm and an inner diameter of 200 μm. The width and depth of the main microchannel were set to be 400 μm and 365 μm, respectively, and those of the side microchannel were 365 μm to accommodate the size of microcapillary. The silica microcapillary was glued further downstream the junction by 90 μm, resulting in a final narrowing width of 60 μm. Specifically, under the microscope, the glue was dripped onto the silica microcapillary that was already located in the designated location of the PMMA microchip with the aid of a 32G microneedle. The glue would rapidly fill up the gap between the silica microcapillary and fix the silica microcapillary in the designated location.

### 2.3. Scanning Electron Microscope (SEM) Imaging

Alginate–gelatin hydrogels prepared with different parameters including gelatin-to-alginate ratio, TGM concentration, and TGM reaction time are summarized in Table 1. Alginate–gelatin pre-polymer solutions of different mass ratios were prepared at 45 °C with the aid of a constantly stirring magnetic bar. The alginate–gelatin precursor solutions were injected into a 6-well plate that served as mold for downstream analysis. To simulate the real-case microfiber fabrication with microfluidic device, the 6-well plate containing alginate–gelatin precursor solution was transferred in a 4 °C refrigerator for 10 min to freeze the gelatin content. Then, 1.5 mL of 1% CaCl_2_ solution was injected into each well to cross-link the alginate content. The CaCl_2_ solution was washed away twice with deionized water, followed by the addition of TGM cross-linking solutions of specific concentration. After the completion of the designated reaction time, the TGM cross-linking solution was washed away twice with deionized water. Deionized water was sipped away after the washing step, and the generated samples were then frozen at −80 °C overnight. Finally, the samples underwent freeze-drying and became ready for SEM imaging. Freeze-dried samples were deposited on a silicon wafer and were gold-coated for 90 s prior the SEM imaging. The SEM imaging was conducted using FEI QUANTA 200.

### 2.4. Assessment for Material Viscoelasticity and Stress–Relaxation

The samples were prepared in a similar fashion as for SEM imaging, except that the hydrogels were not frozen at −80 ℃ nor then freeze-dried. The material viscoelasticity and stress–relaxation were assessed using an Anton Paar MCR301 dynamic rheometer (Graz, Austria). The viscoelasticity of the hydrogel was measured at the strain of 10% and over the angular speed ranging from 1 to 1000 rad per second. The stress–relaxation of the hydrogel was measured at the strain of 10% and over a period of 300 s, and the data were fit according to the Maxwell model with *n* elements as described in Equation (1).
(1)$$\sigma =\sum_{n}\sigma_{n}e^{-\frac{t}{\tau_{n}}}\;$$
where σ is the shear stress in Pa, *it* is time in s, and τ is the relaxation time in s.

### 2.5. Assessment for Material Stiffness

To characterize the stiffness of the alginate–gelatin hydrogel, we used an mmi Cell manipulator optical trapping system (mmi Cellmanipulator, MM1 AG, Zurich, Switzerland) [61] to determine the viscoelasticity of alginate–gelatin hydrogel. In the experiment, polystyrene beads (d = 5 μm, Hugebio, Shanghai, China) were added to the surface of the hydrogel and trapped by the laser moving along a certain direction at a constant oscillation velocity. The stiffness of the hydrogel was calculated by the slope of the force-distance curve [62].

### 2.6. Assessment for Material Water Absorption Ability

The samples were prepared in a similar fashion as for SEM imaging. The dry weight of each sample was measured and compared with its corresponding mass after three days of immersion in water at 37 °C inside the incubator to fully simulate the cultivation conditions. The water absorption ability is defined as $\%\mathrm{water}\;\mathrm{uptake}=\frac{\mathrm{Wet}\;\mathrm{weight}}{\mathrm{Dry}\;\mathrm{weight}}\times 100\%$.

### 2.7. Generation of Alginate–Gelatin Microspheres

All experiments were performed at room temperature and atmospheric pressure. The continuous oil liquid and the dispersed alginate–gelatin precursor solution were introduced via polytetrafluoroethylene (PTFE) tubing by two syringe pumps (LSP01-1B, Baoding Longer Precision Pump Co., Ltd. Baoding, China) into the main microchannel and the side microchannel, respectively. The generated microdroplets were collected through a PTFE tubing inserted into a 50 mL tube filled up with 1% CaCl_2_ solution and put on ice to induce the freezing of the gelatin content. The collected microdroplets were then centrifuged at 1000× *g* rpm for 5 min, where the alginate–gelatin microspheres transferred from oil phase into 1% CaCl_2_ solution, and the alginate content cross-linked under the effect of calcium ions. The 1% CaCl_2_ solution was washed away and replaced with TGM cross-linking solution to cross-link covalently the gelatin content of generated microspheres. The resulted microspheres were observed under microscope and measured for their diameters. At least 30 microdroplets were measured per flowrate condition using ImageJ, and the coefficient of variations (CV) were calculated accordingly.

### 2.8. HepG2 Cell Culture

HepG2 (human liver cancer cell line) was purchased from ATCC. HepG2 was cultured on 2D flasks in growth medium: Dulbecco’s modified Eagle’s medium (DMEM, Multicell) supplemented with 10% FBS and 1% PS. Medium was refreshed every 2 days, and cells were passaged before 80% confluence.

### 2.9. Cell Microencapsulation in Alginate–Gelatin Microspheres

The alginate–gelatin precursor solution was firstly sterilized by pouring the solution over a 25-cm Petri dish and exposing the Petri dish under UV light inside the biosafety cabinet (BSC) for 4 h. The microfluidic chip was sterilized via ethylene oxide. HepG2 cells were collected from culture flask, suspended in DMEM culture medium suspension, and mixed with alginate–gelatin precursor solution. The cell-laden microspheres were prepared as described in the above section. However, both the 1% CaCl_2_ solution and TGM cross-linking solution were made with DMEM culture medium as solvent to ensure the cell viability during the microsphere preparation. Notably, both the CaCl_2_ solution and TGM solution were washed twice with fresh DMEM culture medium to remove any residual cross-linker for both alginate and gelatin contents.

### 2.10. Cell Viability and Urea Secretion

To assess the long-term cell viability and proliferation within the biomaterial, cell-laden hydrogel was incubated with Calcein AM and propidium iodide dye (Wako, at 1:1000 dilution in DMEM) at 37 °C for 90 min. Samples were imaged with Nikon Eclipse fluorescent microscope (Tokyo, Japan).

To assess the urea secretion of HepG2 cells encapsulated inside microgels, the urea detection kit from Bestbio Inc. (Nanjing, China) was used. Briefly, the kit employs the diacetyl monoxime method to determine the urea content in the culture medium of cell-laden microgels.

## 3. Results and Discussion

As shown in Figure 1, by utilizing the natural property of gelatin to rapidly freeze under coldness, cell-laden alginate–gelatin microspheres can undergo a pre-gelation process in the oil phase on ice and maintain their morphologies when being centrifuged into the aqueous culture medium, thereby maximizing the throughput of the microsphere collection. Without any molecule functionalization nor addition of other chemicals, the microsphere morphologies and the maximum cell viability inside hydrogel can be ensured. After alginate–gelatin microspheres are centrifuged into the CaCl_2_-containing cell culture medium, calcium ions would cross-link the alginate content that later serves as the template to maintain the shape of microspheres when the gelatin content melts in the incubator. To further stabilize the gelatin content of cell-laden microspheres, TGM is added to covalently cross-link the gelatin part of hydrogel.

As abovementioned, in this work, the viscoelasticity of the alginate–gelatin network can be finely tuned over a wide range under the combined effect of ionically cross-linked alginate and covalently cross-linked gelatin by modulating parameters including the gelatin-to-alginate ratio, TGM concentration, and TGM reaction time. The alginate and gelatin content also interact with each other via the electrostatic forces between the amine group of gelatin and hydroxyl groups of alginate, as well as the H-bonding among the hydroxyl groups of both components. Here, the viscoelasticity, stiffness, stress–relaxation, and water absorption ability of alginate–gelatin hydrogels made of different gelatin-to-alginate ratios (from 3.75 to 15), TGM concentrations (from 5 to 20 mg/mL), and TGM reaction times (from 30 to 120 min) are summarized in Figure 3.

As shown in Figure 3A, when raising the TGM concentration and/or extending the TGM reaction time for hydrogels of all gelatin-to-alginate ratios, the storage modulus G′ (Young’s modulus) would increase as more gelatin content was cross-linked within the co-polymer network under the action of TGM, thereby enhancing the mechanical strength of the alginate–gelatin hydrogel. However, when raising the TGM to 20 mg/mL for hydrogel with the gelatin-to-alginate ratio of 15, the prolongation of the TGM reaction time from 60 min to 120 min would result in a decrease in the value of G′. This might be due to the swelling of hydrogel originated from the prolongation of the immersion time, leading to the loss in hydrogel strength [63]. When the TGM concentration is low, the TGM could penetrate inward the hydrogel along with its immersion in the cross-linking solution, leading to the gelation of more gelatin content and enhancing the hydrogel strength. When the TGM concentration is too high, the hydrogel network would become overly condensed and prevent the further penetration of TGM inward of the hydrogel. Therefore, less gelatin is cross-linked throughout the hydrogel, leading to a weaker storage modulus G′. However, as illustrated in Figure 3A, the result of viscoelasticity showcased that the gelation was relatively difficult for hydrogel with a gelatin-to-alginate ratio of 7.5, especially when the TGM concentration was low and/or the TGM reaction time was kept short. In fact, for hydrogel with a gelatin-to-alginate ratio of 3.75, the high alginate content quickly cross-linked under the action of calcium ions, formed the essential backbone for the co-polymer, and restrained the diffusion of melted gelatin into the surrounding environment. For hydrogel with a gelatin-to-alginate ratio of 15, the low alginate content could not create much spatial hinderance for the cross-linking of gelatin, allowing the efficient gelation of the gelatin content under the action of TGM. On the other hand, at the gelatin-to-alginate ratio of 7.5, while the alginate content could not quickly form the backbone network supporting the hydrogel and restrain the diffusion of melted gelatin, it could still create a significant spatial hinderance hampering the cross-linking of gelatin content, thereby preventing the efficient gelation of the as-designed co-polymer. Overall, when decreasing the gelatin-to-alginate ratio from 15 to 3.75, the storage modulus G’ would raise when keeping other parameters identical. When looking at the damping factor theta of the alginate-gelatin hydrogel where a higher value signifies a more “liquid-like” hydrogel and a lower value represents a more “solid-like” hydrogel, the results conform with our previous explanation. The damping factor of the alginate-gelatin hydrogel decreased along with the raise of the TGM reaction time and/or the TGM concentration for all gelatin-to-alginate ratios, meaning that the hydrogel turned from more “liquid-like” into more “solid-like” with the progression of gelatin cross-linking. Such a decrease was more significant at the high gelatin-to-alginate ratio since the low alginate content could not spatially hinder the gelation of gelatin under the action of TGM. The damping factor also decreased along with the gelatin-to-alginate ratio since the alginate content with high rigidity would shift the hydrogel into a more “solid-like” state. However, for hydrogel with the gelatin-to-alginate ratio of 15, when the TGM concentration was set to be 20 mg/mL, the extension of the TGM reaction time would raise the damping factor, shifting the hydrogel into a more “liquid-like” state. As previously described, because the low alginate content would not spatially hinder the cross-linking of gelatin under the action of TGM, a too high TGM concentration would quickly form a condense hydrogel network, preventing its further penetration inward of the hydrogel. Instead, with the immersion in the cross-linking solution, the hydrogel would swell and turn into “liquid-like” state. Notably, the abnormalities at the TGM concentration of 5 mg/mL and the TGM reaction time of 30 min for hydrogel of all gelatin-to-alginate ratios were due to the bad gelation at such a low cross-linker concentration as well as cross-linking time.

Similarly, as shown in Figure 3C, the stress–relaxation of the alginate–gelatin hydrogel, where a higher relaxation time signifies a more “solid-like” hydrogel, conformed with the results of the viscoelasticity. Overall, the raise in both the TGM concentration and/or TGM reaction time would enhance the cross-linking of gelatin and density of hydrogel, resulting in a more “solid-like” co-polymer. Likewise, the less efficient gelation at the gelatin-to-alginate ratio of 7.5 led to a more “liquid-like” stress–relaxation behavior, showcased by the overall smaller relaxation time. On the other hand, the stiffness of alginate–gelatin hydrogel measured through optical tweezers further confirmed our previous analysis over the hydrogel mechanical performance, as the raise in both the TGM concentration and/or TGM reaction time enhanced the stiffness of alginate–gelatin hydrogel, as illustrated in Figure 3B. In addition, the water uptake ability of the alginate–gelatin hydrogel shown in Figure 3D is a key parameter to determine the water affinity and the biocompatibility of hydrogel as it would determine the ease of aqueous-based nutrient, oxygen, and waste transport across cell-laden hydrogel. While the water uptake ability showcased a decrease when increasing the TGM cross-linking time and TGM concentration because of the generation of more condense hydrogel, the water uptake ability is strongly dependent on the alginate content of the scaffolds since alginate has a higher water affinity when compared to gelatin. The variation in the water uptake ability is also related to that in hydrogel stiffness, as a more cross-linked hydrogel would have a stronger stiffness as well as water absorption ability, which conforms with previous work [64].

To better visualize the microscopic structure of the generated alginate–gelatin hydrogel at different conditions, the SEM images were taken and illustrated in Figure 4. As expected, the macroporous structure of the hydrogel was strongly depended on the alginate–gelatin ratio: the lower the alginate content, the more porous the scaffold microstructure under SEM. For instance, when compared hydrogels made of the gelatin-to-alginate ratio of 3.75, 7.5, and 15, when keeping all other parameters identical, the hydrogel consisting of 0.5% alginate and 7.5% gelatin showed great porosity since the abundant amount of gelatin when compared to that of alginate effectively hindered the gelation of alginate content and thereby avoiding the emergence of large surface of condense hydrogel. Similarly, when the gelatin cross-linking time was kept relatively short, i.e., lower than 60 min, the scaffold microstructure was also very porous since a minimal amount of gelatin was cross-linked within such a short reaction time. Under the action of freeze-drying, non-cross-linked gelatin was evaporated away, leaving the porous microstructure in place. This would also suggest a relatively weak mechanical strength, as demonstrated by Figure 3. When further increasing the alginate content or the gelatin cross-linking time, as shown in Figure 4, the pore size would increase, and the microstructure would gradually shift from porous to sheet-like because more alginate would be cross-linked and would form a stronger network encapsulating gelatin content.

In general, the results shown in Figure 3 and Figure 4 showcased that the as-designed alginate–gelatin co-polymer hydrogel exhibited a tunable viscoelastic mechanical performance as well as microstructure of varying porosity and pore size over a wide range depending on different cross-linking parameters including gelatin-to-alginate ratio, TGM concentration, and TGM reaction time. Hence, the biophysical microenvironment of the alginate-gelatin hydrogel can be easily modulated and optimized for the specific encapsulated cell type, which would maximize both their proliferation, differentiation, and functionalities. The co-polymer hydrogel system consisting of 1% alginate and 7.5% gelation along with a TGM concentration of 10 mg/mL and a TGM cross-linking time of 60 min was chosen to further illustrate the microfluidic-based cell microencapsulation of the designed system.

From a hydrodynamic perspective, the microfluidic emulsification often faces the dilemma between microdroplet monodispersity and high throughput, which are both greatly appreciated to ensure the homogeneous growth conditions across microgels and the commercialization of the technology, respectively. In fact, squeezing or dripping regime can generate monodisperse microdroplets at the cost of low throughput and high device cost if a small microdroplet is desired as its size heavily relies on that of the microchannel; alternatively, despite its high throughput and ability to overcome the size limitation of the microchannel, a jetting regime would usually produce a microdroplet of great polydispersity. Such a problem would be further emphasized when using a biopolymer precursor solution of high viscosity as dispersed liquid, which inevitably raises its capillary number (*Ca*) [65,66,67]. Hence, the dilemma of generating monodisperse and small microdroplets at high throughput for the real-life applications remains to be solved. Different approaches have been reported over past years to solve such an issue. For instance, by applying a non-uniform magnetic field onto the ferrofluid inside a T-shaped microchannel, one can accurately tune the applied force filed gradient and control the rapid microdroplet breakup [68]. Here, with the use of the novel step-T-junction microchannel with embedded microcapillary that concentrates the shearing force at the special narrowing zone, monodisperse alginate–gelatin whose size overcomes the dimensional limitation of the microchannel itself could be generated at a high throughput jetting regime while maintaining a CV below 5%, i.e., limit of monodispersity, as shown in Figure 5A [60,69,70,71]. In fact, as suggested by the results of the numerical simulation from our previous work, the very special microstructure of the modified step-T-junction microchannel, which is characterized in the serial narrowing, expansion, and re-narrowing regions from the left wall of the microcapillary to the inside of the microcapillary to the narrowing region, respectively, would bring a double disturbance onto the fluids and consequently facilitate the microdroplet pinch-off [60]. The maintenance of the stable jetting regime generating monodisperse small microdroplets at high throughput relies on the competition between the stability of the dispersed thread dominated by the inertia and viscosity of dispersed liquid, and the disturbance caused by the continuous viscous force as well as interfacial tension. When increasing the viscosity of dispersed biopolymer precursor solution or raising the surfactant concentration in the continuous liquid, the O/W interface would be stabilized, enabling the emergence of stable jetting regime. However, above a certain critical value, the stabilization of the O/W interface would lead to an overly long dispersed thread, allowing the occurrence of multiple pressure-difference spots under the effects of disturbance brought about by the continuous viscous force and interfacial tension and resulting in an unstable jetting regime.

The microscopic image of Figure 5B proved the workability of the “dual-template” method to generate an alginate–gelation microsphere: the frozen gelatin content first acts as the template for the gelation of alginate content under the action of CaCl_2_, and the cross-linked alginate would then serve as the template for the cross-linking of gelatin with the addition of TGM. The resulting alginate–gelatin microspheres maintained a good monodispersity and sphericity. As illustrated by Figure 5C, the average diameters of generated alginate–gelatin microdroplets were determined over the dispersed flowrate ranging from 20 μL/min to 100 μL/min and the continuous flowrate from 200 μL/min to 1000 μL/min. Overall, the average microdroplet diameter decreased with the decrease in disperse flowrate and the raise in continuous flowrate, which conformed with previous findings. The diameters of generated microdroplets ranged from 278 μm to 67 μm, which was one order of magnitude smaller than the microchannel diameter of 400 μm, thereby overcoming the dimensional limitation of the microchannel itself. Moreover, the results of Figure 5D,E suggested that the generation of monodisperse alginate–gelatin microdroplets whose CV was below 5% limit at high throughput was possible over a large range of flowrate conditions, further proving the applicability of the designed microfluidic device in the real-life biomedical application at a large scale.

To demonstrate the biocompatibility of the designed microfluidic system, HepG2 cells were encapsulated inside the alginate–gelatin microspheres over a period of 6 days. As described in the experimental section, HepG2 cells were suspended in the dispersed alginate–gelatin solution with a final concentration of approximately 2.4 × 10^6^ cells per mL. By setting the dispersed flowrate to 60 μL/min and the continuous flowrate to 800 μL/min, the microdroplet generation rate was calculated to be around 1228 microdroplets per second. Together, the HepG2 cell encapsulation rate was estimated to be 2400 cells per second, with a final concentration of around two cells per alginate–gelatin microsphere. As shown in Figure 6A, the live/dead staining images of cell-laden microspheres suggested that the cell viability was maintained over a long-term cultivation of 6 days, and cell proliferation may occur considering the enhancing in the fluorescence signal. This demonstrates that utilizing the natural property of gelatin to freeze under coldness to transfer alginate–gelatin microspheres from cytotoxic oil liquid into aqueous cell culture medium enables the generation of cell-laden alginate–gelatin microspheres in a biocompatible and highly efficient manner. Furthermore, the urea secretion of encapsulated cells, which is an inherent ability of the HepG2 human liver cancer cell line, was tested by measuring the urea content in the culture medium via the diacetyl monoxime method. The results of Figure 6B showcase the existence of urea in the culture medium, and it increased along with the cultivation of cell-laden microspheres, suggesting the maintenance of cell normal functionalities inside the cell-laden microgels and indirectly proofing the proliferation of encapsulated cells.

## 4. Conclusions

Overall, generating microdroplets of desired morphologies at low cost and high throughput is essential for both laboratory and industrial usages. In this work, a novel step-T-junction microchannel with embedded microcapillary was employed to generate monodisperse cell-laden alginate–gelatin microspheres at high throughput. The novel microchannel geometry overcomes the dilemma of microfluidic emulsification between the monodispersity of generated microgels and the throughput of the system. Because of the special dual-disturbance design at the step-T-junction, a stable jetting regime can be achieved with viscous dispersed and continuous liquids, enabling the generation of microdroplets with CV below 5% at high throughput. Depending on the flowrate conditions, the diameter of generated microdroplets could range from 278 to 67 μm, overcoming the dimensional limitation of microchannel and consequently lowering the fabrication cost of miniature microfluidic device. Moreover, by modulating different parameters of alginate–gelatin hydrogel including the gelatin-to-alginate ratio, TGM concentration, and TGM reaction time, the biophysical properties of microgels such as viscoelasticity, stress–relaxation, stiffness, and water uptake ability can be fully modulated to accommodate the specific requirement of the encapsulated cell type and maximize the cell spreading, proliferation, differentiation, and functionalities. The long-term cultivation of HepG2 cell-laden alginate–gelatin microspheres showed a good cell viability over a period of 6 days, proofing the excellent biocompatibility of the “dual-template” method of de-emulsification and gelation. Encapsulated HepG2 cells maintained their proper functionalities of secreting urea. All these results suggest the great potential of the designed microfluidic system with the alginate–gelatin co-polymer network in offering a novel pathway for the real-life applications of cell-laden microgels in tissue engineering, cell therapy, and other biomedical-related uses.

## Figures and Tables

**Figure 1 biosensors-12-00659-f001:**
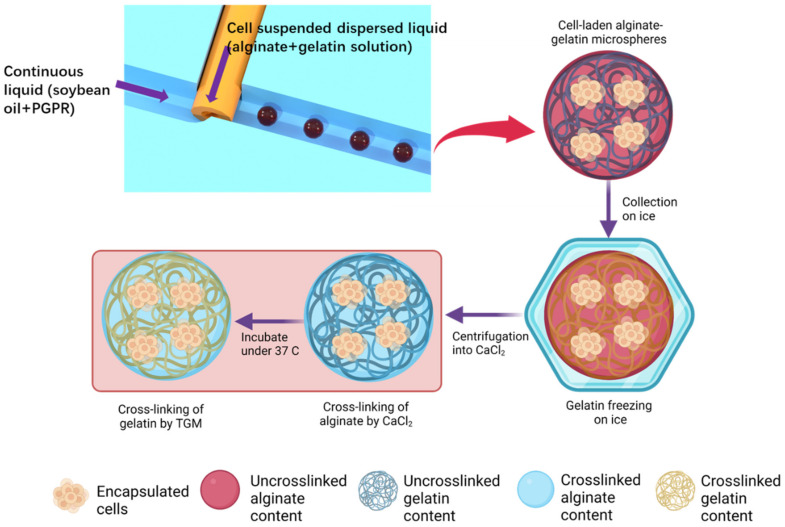
The schematic illustration of the microfluidic cell encapsulation system employing the novel step-T-junction microchannel with embedded microcapillary: the cell-laden alginate–gelatin microdroplets were collected on ice, allowing the freezing of the gelatin content under coldness; then, the alginate–gelatin microspheres were centrifuged from oil continuous liquid into cell culture medium containing CaCl_2_ and cross-linking the alginate content of the microspheres; eventually tranglutaminase (TGM) was added to enable the secondary gelation of the gelatin content.

**Figure 2 biosensors-12-00659-f002:**
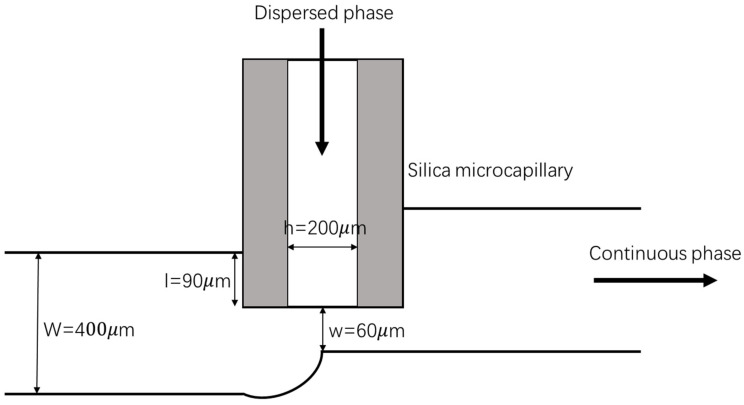
The 2D schematic illustration of the step-T-junction microchannel with embedded microcapillary.

**Figure 3 biosensors-12-00659-f003:**
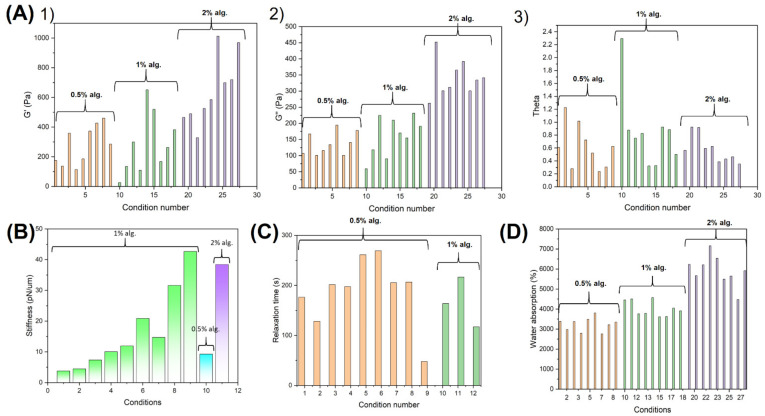
Adjustable mechanical properties of alginate–gelatin hydrogel: (**A**) viscoelasticity (at 1 Hz) of alginate–gelatin hydrogel prepared at different conditions where the condition numbers correspond to those indicated in Table 1: (**1**) storage modulus G′ (Young’s modulus); (**2**) loss modulus G″; (**3**) damping factors of alginate–gelatin hydrogel. (**B**) Stiffness of alginate–gelatin hydrogel at different conditions where the conditions number for hydrogel made of 1% alginate correspond to those indicated in Table 1, and the hydrogels consisted of 0.5% alginate and 1% alginate were made with 10 mg/mL of TGM and 60 min of TGM reaction time. (**C**) Stress–relaxation time ($\tau_{\frac{1}{2}}$) of alginate-gelatin hydrogel at different conditions where the conditions number for hydrogel made of 0.5% alginate correspond to those indicated in Table 1, and the hydrogels consisting of 1% alginate were made with 10 mg/mL of TGM and 30, 60, and 120 min of TGM reaction time. (**D**) Water absorption of alginate–gelatin hydrogel at different conditions.

**Figure 4 biosensors-12-00659-f004:**
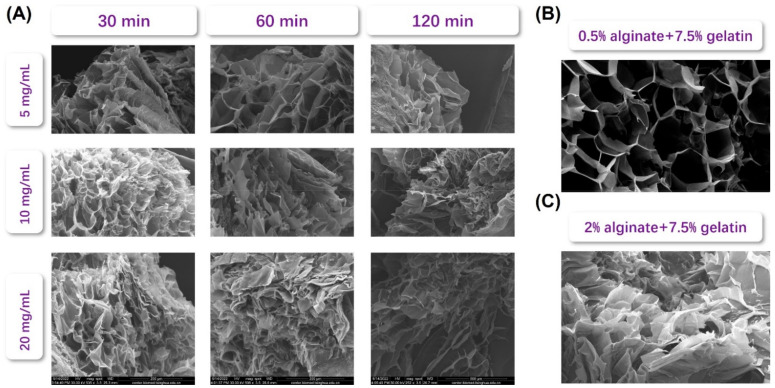
Microscopic structures of alginate–gelatin hydrogel illustrated by SEM images: (**A**) microscopic structure of hydrogel made of 1% alginate and 7.5% gelatin, when the TGase concentration varied from 5 to 20 mg/mL, and the TGase reaction time from 30 to 120 min; (**B**) microscopic structure of hydrogel made of 1% alginate and 7.5% gelatin with TGase concentration of 10 mg/mL and TGase reaction time of 120 min. (**C**) microscopic structure of hydrogel made of 2% alginate and 7.5% gelatin with TGase concentration of 10 mg/mL and TGase reaction time of 120 min.

**Figure 5 biosensors-12-00659-f005:**
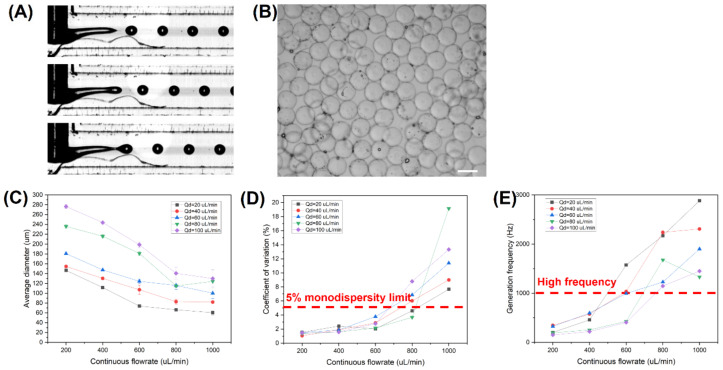
Microfluidic-based fabrication of alginate–gelatin microspheres: (**A**) generation of monodisperse alginate–gelatin microspheres at high-throughput stable narrowing jetting regime; (**B**) microscopic images of typical alginate–gelatin microspheres generated by the designed microfluidic system, wherein the scale bar is 150 μm; (**C**) average microdroplet diameters at different flowrate conditions with dispersed liquid of 1% alginate +7.5% gelatin and continuous phase of soybean oil +8% PGPR; (**D**) CV of microdroplets generated at different flowrate conditions; (**E**) microdroplet generation frequency at different flowrate conditions.

**Figure 6 biosensors-12-00659-f006:**
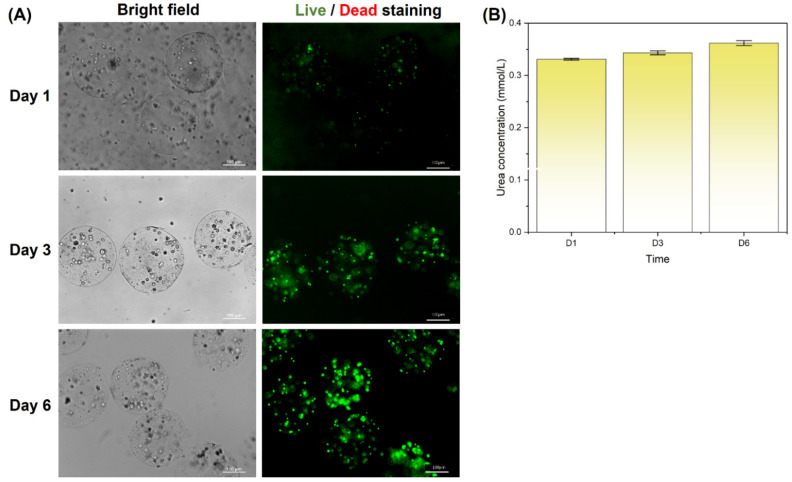
Long-term cellular viability and functioning inside alginate-gelatin microspheres: (**A**) live/dead staining of HepG2 cells encapsulated inside alginate–gelatin microspheres at days 1, 3, and 6 post-encapsulation. (**B**) Urea secretion of HepG2 cells encapsulated inside alginate–gelatin microspheres at days 1, 3, and 6 post-encapsulation.

**Table 1 biosensors-12-00659-t001:** Different parameters affecting the mechanical properties of alginate–gelatin hydrogel and tested in the present study.

No.	Alginate Concentration (wt %)	Gelatin Concentration (wt %)	TGM Concentration (mg/mL)	TGM Reaction Time (min)
1	0.5	7.5	5	30
2	0.5	7.5	5	60
3	0.5	7.5	5	120
4	0.5	7.5	10	30
5	0.5	7.5	10	60
6	0.5	7.5	10	120
7	0.5	7.5	20	30
8	0.5	7.5	20	60
9	0.5	7.5	20	120
10	1	7.5	5	30
11	1	7.5	5	60
12	1	7.5	5	120
13	1	7.5	10	30
14	1	7.5	10	60
15	1	7.5	10	120
16	1	7.5	20	30
17	1	7.5	20	60
18	1	7.5	20	120
19	2	7.5	5	30
20	2	7.5	5	60
21	2	7.5	5	120
22	2	7.5	10	30
23	2	7.5	10	60
24	2	7.5	10	120
25	2	7.5	20	30
26	2	7.5	20	60
27	2	7.5	20	120

## Data Availability

Not applicable.

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
