# Peer review of "A Novel Step-T-Junction Microchannel for the Cell Encapsulation in Monodisperse Alginate-Gelatin Microspheres of Varying Mechanical Properties at High Throughput"

_biosensors, 2022, doi:10.3390/bios12080659_

Round 1
Reviewer 1 Report
See the attached file.

Author Response
Title: “A novel step T-junction microchannel for the cell encapsulation in monodisperse alginate-gelatin microspheres of varying mechanical properties at high throughput”
Manuscript # Biosensors-1851843
Dear Editors and Reviewers,
We greatly appreciate your generous and constructive comments on the manuscript “A novel step T-junction microchannel for the cell encapsulation in monodisperse alginate-gelatin microspheres of varying mechanical properties at high throughput” (# Biosensors-1851843). Your comments are of great values towards our manuscript. We have carefully re-evaluated the manuscript and edited it to address your concerns.
In particular, the encapsulation rate of HepG2 was calculated and added to the revised manuscript to illustrate the performance of the as-designed microfluidic system to encapsulate cells in a quantitative manner. Additional descriptions on the fabrication of microfluidic system and explanations about the fluid mechanics in modulating the stability of the jetting regime were added in the revised Manuscript, respectively. Lastly, the language quality of the writing was further polished in the revised Manuscript.
Please refer to the attached document for detail information about the editing and changes made on the manuscript.
We look forwards to hearing from you regarding our revised manuscript. We would be glad to respond to any further question, concern and comment that you may raise.
Thank you again for your time and effort in editing and reviewing the manuscript, we are very grateful for your work.
Sincerely yours,
Si Da Ling

Reviewer 2 Report
The authors designed a novel step-T-junction microchannel with embedded microcapillary for generating monodisperse cell-laden alginate-gelatin microspheres at high-throughput level. By optimizing the stiffness, stress-relaxation and viscoelasticity, the alginate-gelatin hydrogel offers a good 3D mechanical microenvironment for encapsulating cancer cells. Totally, the work is interesting, the datas basically can support the idea. However, minor revision is needed before it can further be accepted.
1. How about the encapsulation rate of the HepG2 cells into alginate-gelatin microspheres ?
2. Figure 5a only show the imaging live/dead staining results of cell-laden microspheres, the authors may further show the quantification cell viability results to intuitively demonstrate the biocompatility of the microfluidic system.
3. The authors may compare the urea secretion of normal and encapsulated HepG2 cells to see if any difference of the inherent ability of encapsulated HepG2 cells.
Author Response

(The authors gave the same response as above.)

Round 2
Reviewer 1 Report
As I mentioned in my previous set of comments that T-junctions are used to yield several fluidic functionalities in small-scale systems. In order to strengthen the novelty in their revised version, the authors should consider the following recent article in the revised version.
Magnetofluidic-based controlled droplet breakup: effect of non-uniform force field, Journal of Fluid Mechanics 944, 2022. DOI:10.1017/jfm.2022.504

Author Response
Title: “A novel step T-junction microchannel for the cell encapsulation in monodisperse alginate-gelatin microspheres of varying mechanical properties at high throughput”
Manuscript # Biosensors-1851843
Dear Editors and Reviewers,
We greatly appreciate your generous and constructive comments on the manuscript “A novel step T-junction microchannel for the cell encapsulation in monodisperse alginate-gelatin microspheres of varying mechanical properties at high throughput” (# Biosensors-1851843). Your comments are of great values towards our manuscript. We have carefully re-evaluated the manuscript and edited it to address your concerns.
In particular, we have carefully read the article proposed by the reviewer, and considered very inspiring to support our work. Therefore, we have described the article and cited it in our revised manuscript.
Please refer to the attached document for detail information about the editing and changes made on the manuscript.
We look forwards to hearing from you regarding our revised manuscript. We would be glad to respond to any further question, concern and comment that you may raise.
Thank you again for your time and effort in editing and reviewing the manuscript, we are very grateful for your work.
Sincerely yours,
Si Da Ling
Response to reviews’ comments
Reviewer’s comments
Comments: As I mentioned in my previous set of comments that T-junctions are used to yield several fluidic functionalities in small-scale systems. In order to strengthen the novelty in their revised version, the authors should consider the following recent article in the revised version.
Magnetofluidic-based controlled droplet breakup: effect of non-uniform force field, Journal of Fluid Mechanics 944, 2022. DOI:10.1017/jfm.2022.504
Response to comment
Thank you for your suggestion. After carefully reading the proposed article, we have added the citation in the revised manuscript.
